# The effect of interleukin 10 polymorphisms on breast cancer susceptibility in Han women in Shaanxi Province

Miao Li[1,2]☯, Chenli Yue[3]☯, Xiaoxiao Zuo[4], Guoquan Jin[5], Guanying Wang[6], Hulin Guo[2], Fang Wu[7], Shangke Huang[8], Xinhan Zhao📷[1]*

1 Department of Medical Oncology, The First Affiliated Hospital of Xi'an Jiaotong University, Xi'an, Shaanxi, China, 2 Department of Internal Medicine Oncology, The Fifth People's Hospital of Qinghai Province, Xining, Qinghai, China, 3 Department of Respiratory Medicine, Shaanxi Provincial Crops Hospital of Chinese Peoples Armed Police Force, Xi'an, Shaanxi, China, 4 Department of Radiation Oncology, First Affiliated Hospital of Zhengzhou University, Zhengzhou, Henan, China, 5 Department of General Surgery, The Fifth People's Hospital of Qinghai Province, Xining, Qinghai, China, 6 Department of Medical Oncology, The Second Affiliated Hospital of Xi'an Jiaotong University, Xi'an, Shaanxi, China, 7 Department of Neonatology, The First Affiliated Hospital of Xi'an Jiaotong University, Xi'an, Shaanxi, China, 8 Department of Oncology, The Affiliated Hospital of Southwest Medical University, Luzhou, Sichuan, China

☯ These authors contributed equally to this work.
* zhaohanxin21@163.com

**Data Availability Statement:** All relevant data are within the manuscript.

**Funding:** The author(s) received no specific funding for this work.

## Abstract

### Background

Previous studies have reported on several genetic variants related to breast cancer, but a substantial proportion of mutation loci have not yet been identified. In the current study, we aimed to evaluate the association between single nucleotide polymorphisms (SNPs) of interleukin-10 (*IL-10*) and susceptibility to breast cancer in Shaanxi Han women in China.

### Methods

Six SNPs were genotyped in 530 breast cancer patients and 628 healthy women from the First Affiliated Hospital of Xi'an Jiaotong University Hospital. Odds ratios and 95% confidence intervals were calculated by unconditional logistic regression analysis to assess the association between breast cancer risk and polymorphisms of six loci.

### Results

Two SNPs, rs3024490 and rs1800871, were found to be significantly different between breast cancer patients and healthy women. These SNPs also increased the risk of breast cancer in co-dominant and dominant models. Moreover, another SNP, rs1554286, was significantly associated with an increased risk of breast cancer in the co-dominant model. Functional assessments indicated that these three variants may influence the expression and transcription factor binding of *IL-10*.

**Competing interests:** The authors have declared that no competing interests exist.

**Abbreviations:** CIs, Confidence intervals; eQTL, Expression quantitative trait loci; ER, estrogen receptor; GTEx, Genotype-tissue expression; GWAS, Genome-wide association study; HER2, human epidermal growth factor receptor 2; HWE, Hardy-Weinberg Equilibrium; IL-10, Interleukin-10; LD, Linkage disequilibrium; MAF, minor allele frequency; ORs, Odds ratios; PR, progesterone receptor; SD, standard deviation; SNPs, Single nucleotide polymorphisms.

## Conclusions

Our findings suggest that variants of *IL-10* may be likelihood risk factors for the development and progression of breast cancer. Future studies should replicate this study and evaluate functional assessments in Chinese Han women and women from other regions.

## Background

Breast cancer is a life-threatening cancer among women and is the sixth most common cause of death among women in China [1]. The American Cancer Society estimated that there were 252,710 new breast cancer cases in the United States in 2017, and approximately 40,610 deaths in women occurred due to breast cancer [2]. Although numerous environmental factors are well known to cause breast cancer, genetic factors have been confirmed to play a crucial role in the progression of the disease [3, 4].

A growing number of genome-wide association studies have found that variants in genes can significantly affect the susceptibility to breast cancer [5, 6]. However, currently identified single nucleotide polymorphisms (SNPs) can only shed partial insight into the inherited causes of breast cancer, and the precise mechanisms underlying breast cancer remain unclear. Therefore, it is valuable to explore novel SNP loci for assessing the risk of breast cancer. The human interleukin-10 gene (*IL-10*) is located on chromosome 1q32.1 and contains four introns and five exons [7]. IL-10 is a highly pleiotropic cytokine that is produced by various cells. Studies have shown that IL-10 plays an important role in the initiation and development of breast cancer [8]. Expression of IL-10 is higher in breast cancer patients than in healthy persons [9, 10]. Ahmad et al. [11] demonstrated that IL-10 has an inhibitory effect on the migration of breast cancer cell lines, and this study also found that breast cancer patients with higher IL-10 levels show better disease-free survival and breast cancer-specific survival in the early stages of invasive breast cancer. Thus, IL-10 has been suggested as a potential prognostic marker for breast cancer [12], and there may be a genetic correlation between *IL-10* and breast cancer susceptibility. Several studies [13, 14] have investigated the relationships between *IL-10* polymorphisms and breast cancer risk, but have not completely clarified.

Here, we genotyped six SNPs (rs1554286, rs1518111, rs3021094, rs3790622, rs3024490, rs1800871) of IL-10 genes in a case-control study to explore the association between polymorphisms of *IL-10* and the risk of breast cancer in Northwest Chinese Han women. Our data shed new light on the association between IL-10 SNPs and breast cancer susceptibility in Shaanxi Han women in China.

## Methods

### Ethics approval and consent to participate

All participants were informed of the procedures and purpose of this study. Signed informed consent documents were obtained from both patients and healthy individuals. Study protocols were approved by the Ethical Committee of the First Affiliated Hospital of Xi'an Jiaotong University, and complied with the ethical standards of the Ethical Committee and World Medical Association Declaration of Helsinki. All subsequent research analyses were carried out in accordance with approved guidelines and regulations.

## Study participants

The case and control study included 530 breast cancer patients and 628 healthy controls, who were recruited from September 2016 to June 2019 at the First Affiliated Hospital of Xi'an Jiaotong University. The case groups confirmed breast cancer through histological examination, and all cases were diagnosed with a new breast cancer event. Patients diagnosed with other types of cancer and / or receiving hormone therapy, radiotherapy or chemotherapy were excluded. The control groups were healthy women who had undergone an annual health assessment and were recruited from a health checkup center affiliated to our institution. These women did not have any positive signs of cancer and had no history of malignant family history. The case and control groups were all Chinese Han women living in Shaanxi Province, and there was no blood relationship between them.

## SNP selection and genotyping

Total genomic DNA was isolated using a GoldMag-Mini Whole Blood Genomic DNA Purification Kit (GlodMag Co. Ltd., Xi'an, China) from peripheral blood, according to the manufacturer's protocol. The Agena Bioscience Assay Design Suite V2.0 software (https://agenacx.com/online-tools/) was used to design the extended primer. SNPs were genotyped using the MassARRAY Nanodispenser (Agena Bioscience, San Diego, CA, USA) and MassARRAY iPLEX platform (Agena Bioscience, San Diego, CA, USA), according to the standard protocol recommended by the manufacturer.

According to the standard protocol recommended by the manufacturer, two laboratory personnel used the MassARRAY Nanodispenser (Agena Bioscience, San Diego, CA, USA) and MassARRAY iPLEX platform (Agena Bioscience, San Diego, CA, USA) to genotype SNPs in a double-blind manner. About 10% of the samples were randomly selected for repeated genotyping to confirm the results, and the reproducibility was 100%. The Agena Bioscience TYPER software (version 4.0) was used for data management and analyses.

In this study, six SNPs (rs1554286, rs1518111, rs3021094, rs3790622, rs3024490, rs1800871) of *IL-10* were selected from the DbSNP (http://www.hapmap.org/index.html.en) and SNP Consortium (http://snp.cshl.org/) databases based on the following criteria. We took allele frequency into consideration during genotyping. The lower frequency alleles were coded as minor alleles. All SNPs were selected at a minor allele frequency >5% in the 1,000 genome project (http://www.internationalgenome.org/). Chinese Han Beijing population, and the genotype distribution of SNPs in the control group was in accordance with Hardy-weinberg equilibrium (HWE) ($p >$ 0.05). The genotyping of Agena MassARRAY RS1000 for these SNPs exceeded 95.0%.

## Statistical analysis

Microsoft Excel and SPSS 18.0 statistical package (SPSS, Chicago, IL, USA) were used for statistical analyses. All $p$-values were based on two-sided tests, and $p \leq 0.05$ was considered as the threshold of statistical significance. Welch's t-test was used to compare ages between cases and controls. The validation of each SNP frequency in control subjects was tested for departure from the Hardy-Weinberg Equilibrium (HWE) using an exact test, and the difference in allele frequency distribution among cases and controls was assessed using Pearson's chi-square test. Subsequently, unconditional logistic regression analysis was used to evaluate breast cancer susceptibility under four genotype models (co-dominant, dominant, recessive, and additive model). The SNPStats software platform (https://www.snpstats.net/start.htm) and Haploview software package (version 4.2) [15] were used to perform the linkage disequilibrium (LD) and analyze the association between haplotypes and the risk of breast cancer. The effects of *IL-10*

polymorphisms on the risk of breast cancer were expressed as odds ratios (ORs) and 95% confidence intervals (CIs) [16].

## Functional annotation

*In silico* analysis of breast cancer-associated SNPs on gene expression was assessed using the genotype-tissue expression (GTEx) database of quantitative trait loci (eQTL) variants. This is used to determine the biological effects of genetic variants (http://www.gtexportal.org). Mapping of eQTLs provides a powerful approach to uncover the genetic factors underlying altered gene expression [17].

Regulome DB (http://www.regulomedb.org/) was used to determine the effect of *IL-10* SNPs on allele-specific transcription factor binding. Regulome DB utilizes chromatin immunoprecipitation-sequencing data and chromatin state information across many cell types, as well as eQTL information, for the functional annotation of variants. The variants are scored based on predicted potential effects caused by the variant residing in a functionally important region of the genome. The lower the score the higher impact on protein binding and expression of the target gene.

In addition, HaploReg (http://pubs.broadinstitute.org/mammals/haploreg/haploreg.php) has been used to explore annotations of the noncoding genome at variants on haplotype blocks [18]. We used HaploReg (version 4.1) to assess the rs3024490- and rs1800871-tagged SNPs using LD information from the 1000 Genomes Project with $r^2 \geq 0.8$.

## Results

### Participant characteristics

A total of 530 breast cancer patients and 628 healthy individuals were enrolled. Demographics and clinical characteristics are shown in Table 1. The mean ages of cases and controls were 50.69 and 51.04 years, respectively. Welch's t-test revealed significant differences in age between cases and controls ($p = 0.001$).

**Table 1. Characteristics of cases and controls.**

| Variables | | Cases (N = 530) | Controls (N = 628) | *p*-value |
|---|---|---|---|---|
| Age, years (mean±SD) | | 50.69 ± 11.74 | 51.04 ± 9.64 | 0.001[a] |
| Age, years | > 50 | 276 (52.08%) | 336 (53.50%) | |
| | ≤ 50 | 254 (47.92%) | 292 (46.50%) | |
| Tumor classification | < 2mm | 232 (43.77%) | | |
| | ≥ 2mm | 298 (56.23%) | | |
| ER status | Negative | 220 (41.51%) | | |
| | Positive | 310 (58.49%) | | |
| PR status | Negative | 278 (52.45%) | | |
| | Positive | 252 (47.55%) | | |
| HER2 status | Negative | 208 (39.25%) | | |
| | Positive | 322 (60.75%) | | |
| Menopause Status | Pre-menopause | 216 (40.75%) | | |
| | Post-menopause | 314 (59.25%) | | |

SD, standard deviation; ER: estrogen receptor; PR, progesterone receptor; HER2, human epidermal growth factor receptor 2.

[a]*p*-value was calculated using Welch's t test.

**Table 2. Allele frequencies in cases and controls and odds ratio estimates for breast cancer.**

| SNP ID | Band | Position | Role | Alleles A/B | MAF | | p-HWE | OR(95%CI) | [a]p-value |
|---|---|---|---|---|---|---|---|---|---|
| | | | | | Case | Control | | | |
| rs1554286 | 1q32.1 | 206944233 | Intron (boundary) | G/A | 0.358 | 0.344 | 0.168 | 1.07(0.84–1.36) | 0.605 |
| rs1518111 | 1q32.1 | 206944645 | Intron (boundary) | C/T | 0.358 | 0.338 | 0.043[#] | 1.1(0.86–1.4) | 0.456 |
| rs3021094 | 1q32.1 | 206944952 | Intron | G/T | 0.436 | 0.447 | 0.022[#] | 0.95(0.76–1.2) | 0.692 |
| rs3790622 | 1q32.1 | 206945163 | Intron | A/G | 0.081 | 0.08 | 0.429 | 1.02(0.67–1.57) | 0.910 |
| rs3024490 | 1q32.1 | 206945311 | Intron | C/A | 0.355 | 0.334 | 0.099 | 1.09(0.86–1.4) | 0.468 |
| rs1800871 | 1q32.1 | 206946634 | Promoter | G/A | 0.355 | 0.334 | 0.099 | 1.09(0.86–1.4) | 0.468 |

SNP, single nucleotide polymorphism; MAF, minor allele frequency; HWE, Hardy-Weinberg Equilibrium; ORs, odds ratios; CI, confidence interval; A, minor allele; B, major allele.

[#]Sites with HWE, $p < 0.05$, are excluded.

[a]$p$-values were calculated using Pearson's Chi-square test.

## Association between IL-10 polymorphisms and breast cancer risk

Detailed SNP data, including the position, allele, and minor allele frequency, are presented in Table 2. Two SNPs, rs1518111 and rs3021094, were not found to be in HWE in the control participants ($p < 0.05$). Therefore, these two SNPs were excluded from subsequent statistical analyses. We assumed the minor allele of SNPs as the risk allele to analyze the correlation with breast cancer susceptibility. Our results show that polymorphisms of all candidate SNPs were not associated with breast cancer risk ($p > 0.05$).

Results from the genotype models are shown in Table 3. Rs1554286 was associated with an increased risk of breast cancer in the co-dominant model (OR = 1.52, 95% CI = 1.07–2.16, $p = 0.018$). After adjusting for age, the OR was 1.51 (95% CI = 1.07–2.15, $p = 0.018$). Rs3024490 was also associated with an increased risk of breast cancer in the co-dominant model (OR = 1.65, 95% CI = 1.16–2.33, $p = 0.004$) and dominant model (OR = 1.43, 95% CI = 1.03–2.00, $p = 0.033$). After adjusting for age, significant associations were found in the co-dominant model (OR = 1.64, 95% CI = 1.16–2.33, $p = 0.004$) and dominant model (OR = 1.43, 95% CI = 1.02–2.00, $p = 0.035$). In addition, the co-dominant model (OR = 1.64, 95% CI = 1.16–2.33, $p = 0.004$) and dominant model (OR = 1.43, 95% CI = 1.02–2.00, $p = 0.035$) showed associations between rs1800871 and an increased risk of breast cancer in multiple comparisons adjusting for age. No significant associations were found for rs3790622.

## Association between haplotypes and breast cancer risk

Finally, LD analysis was used to assess the association between haplotypes and breast cancer susceptibility. Four SNPs (rs1554286, rs3790622, rs3024490 and rs1800871) had a strong LD, and the D' value was 1. This indicated that these four SNPs tended to be co-inherited (Fig 1). Furthermore, the haplotypes were not significantly associated with breast cancer susceptibility (Table 4).

## Functional assessment of breast cancer-associated SNPs on IL-10 expression

After the association study, the effects of rs1554286, rs3024490, and rs1800871 on *IL-10* expression were evaluated using the GTEx database (Table 5). We found that these three SNPs

**Table 3. Association between dominant single nucleotide polymorphisms (SNPs) and the risk of breast cancer in multiple inheritance models (adjusted by age).**

| SNP_ID | Model | Genotype | Genotype Frequencies | | Without adjustment | | With adjustment | |
|---|---|---|---|---|---|---|---|---|
| | | | Case | Control | OR(95%CI) | *p*a | OR(95%CI) | *p*b |
| rs1554286 | Co-dominant | A/A | 200 (37.7%) | 282 (44.9%) | 1 | 0.018* | 1 | 0.018* |
| | | A/G | 280 (52.8%) | 260 (41.4%) | 1.52 (1.07–2.16) | | 1.51 (1.07–2.15) | |
| | | G/G | 50 (9.4%) | 86 (13.7%) | 0.82 (0.47–1.43) | | 0.82 (0.47–1.42) | |
| | Dominant | A/A | 200 (37.7%) | 282 (44.9%) | 1 | 0.081 | 1 | 0.084 |
| | | A/G-G/G | 330 (62.3%) | 346 (55.1%) | 1.34 (0.96–1.88) | | 1.34 (0.96–1.87) | |
| | Recessive | A/A-A/G | 480 (90.6%) | 542 (86.3%) | 1 | 0.11 | 1 | 0.11 |
| | | G/G | 50 (9.4%) | 86 (13.7%) | 0.66 (0.39–1.11) | | 0.65 (0.39–1.10) | |
| | Log-additive | - - - | - - - | - - - | 1.07 (0.84–1.36) | 0.6 | 1.07 (0.83–1.36) | 0.62 |
| rs3790622 | Co-dominant | G/G | 444 (84.1%) | 267 (85%) | 1 | 0.62 | 1 | 0.62 |
| | | A/G | 82 (15.5%) | 88 (14%) | 1.12 (0.71–1.78) | | 1.12 (0.71–1.78) | |
| | | A/A | 2 (0.4%) | 6 (1%) | 0.40 (0.04–3.88) | | 0.40 (0.04–3.90) | |
| | Dominant | G/G | 444 (84.1%) | 534 (85%) | 1 | 0.76 | 1 | 0.75 |
| | | A/G-A/A | 84 (15.9%) | 94 (15%) | 1.07 (0.68–1.69) | | 1.08 (0.68–1.69) | |
| | Recessive | G/G-A/G | 526 (99.6%) | 622 (99%) | 1 | 0.39 | 1 | 0.39 |
| | | A/A | 2 (0.4%) | 6 (1%) | 0.39 (0.04–3.81) | | 0.40 (0.04–3.83) | |
| | Log-additive | - - - | - - - | - - - | 1.02 (0.67–1.57) | 0.91 | 1.03 (0.67–1.57) | 0.91 |
| rs3024490 | Co-dominant | A/A | 200 (37.7%) | 292 (46.5%) | 1 | 0.004* | 1 | 0.004* |
| | | A/C | 284 (53.6%) | 252 (40.1%) | 1.65 (1.16–2.33) | | 1.64 (1.16–2.33) | |
| | | C/C | 46 (8.7%) | 84 (13.4%) | 0.80 (0.45–1.41) | | 0.80 (0.45–1.41) | |
| | Dominant | A/A | 200 (37.7%) | 292 (46.5%) | 1 | 0.033* | 1 | 0.035* |
| | | A/C-C/C | 330 (62.3%) | 336 (53.5%) | 1.43 (1.03–2.00) | | 1.43 (1.02–2.00) | |
| | Recessive | A/A-A/C | 484 (91.3%) | 544 (86.6%) | 1 | 0.072 | 1 | 0.071 |
| | | C/C | 46 (8.7%) | 84 (13.4%) | 0.62 (0.36–1.05) | | 0.61 (0.36–1.05) | |
| | Log-additive | - - - | - - - | - - - | 1.10 (0.86–1.40) | 0.46 | 1.09 (0.86–1.40) | 0.47 |
| rs1800871 | Co-dominant | A/A | 200 (37.7%) | 292 (46.5%) | 1 | 0.004* | 1 | 0.004* |
| | | A/G | 284 (53.6%) | 252 (40.1%) | 1.65 (1.16–2.33) | | 1.64 (1.16–2.33) | |
| | | G/G | 46 (8.7%) | 84 (13.4%) | 0.80 (0.45–1.41) | | 0.80 (0.45–1.41) | |
| | Dominant | A/A | 200 (37.7%) | 292 (46.5%) | 1 | 0.033* | 1 | 0.035* |
| | | A/G-G/G | 330 (62.3%) | 336 (53.5%) | 1.43 (1.03–2.00) | | 1.43 (1.02–2.00) | |
| | Recessive | A/A-A/G | 484 (91.3%) | 544 (86.6%) | 1 | 0.072 | 1 | 0.071 |
| | | G/G | 46 (8.7%) | 84 (13.4%) | 0.62 (0.36–1.05) | | 0.61 (0.36–1.05) | |
| | Log-additive | - - - | - - - | - - - | 1.10 (0.86–1.40) | 0.46 | 1.09 (0.86–1.40) | 0.47 |

SNP, single nucleotide polymorphism; OR, odds ratio; 95% CI, 95% confidence interval.

[a]*p*-values were calculated using unconditional logistic regression analysis.

[b]*p*-values were calculated using unconditional logistic regression analysis with adjustments for age.

*$p \leq 0.05$ indicates statistical significance.

variants significantly affected the expression of *IL-10* in whole blood. Furthermore, Regulome DB was used to annotate the possible functional effect. Rs1554286 likely affected eQTL + TF binding / DNase peak and had a Regulome DB score of 1f, which is classified as "Likely to affect binding and linked to expression of a gene target". Rs3024490 had a Regulome DB score of 6, which is classified as "minimal binding evidence". In addition, rs1800871 was likely to affect TF binding + any motif + DNase peak, and it had a Regulome DB score of 3a, which is classified as "less likely to affect binding".

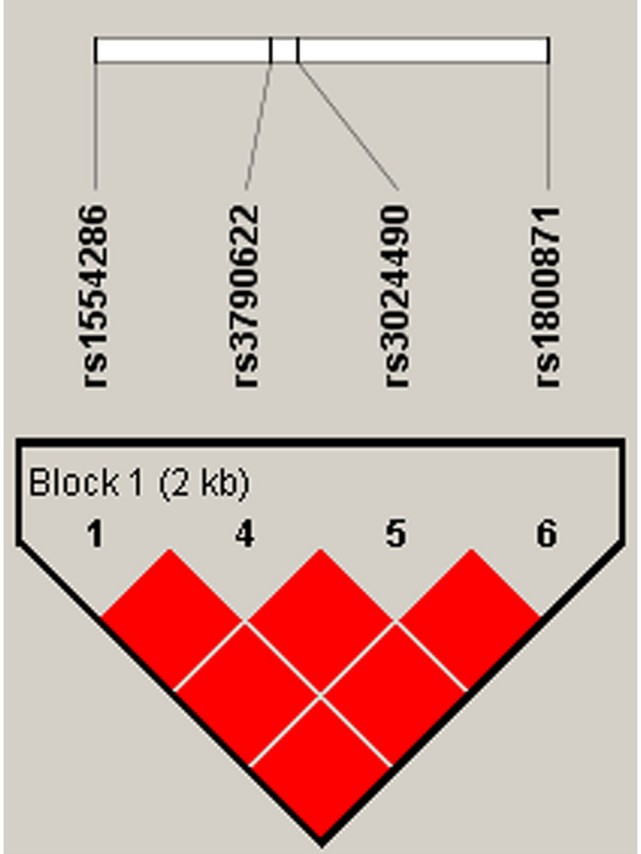

**Fig 1. Haplotype block map for part of the (SNPs) in the IL-10 gene.** Four SNPs in the haplotype map (rs1554285, rs3790622, rs3024490, and rs1800871) had linkage disequilibrium (LD). A standard color frame is used to show the LD pattern. The D value was 1. These four SNPs tended to be co-inherited.

Moreover, we used the HaploReg database to get three genetic variants tagged by the rs3024490 and rs1800871 with $r^2 \geq 0.8$. These three polymorphisms were located in *IL-10* or upstream of *IL-10*. Detailed data, including LD information and predicted function, on these variants are shown in Table 6. These results indicated the functional importance of rs1554286, rs3024490, and rs1800871, which were associated with breast cancer.

**Table 4. *IL-10* haplotype frequencies and the association with breast cancer risk.**

| Haplotype | rs1554286 | rs3790622 | rs3024490 | rs1800871 | Freq-case | Freq-control | Without adjustment | | With adjustment | |
|---|---|---|---|---|---|---|---|---|---|---|
| | | | | | | | OR(95%CI) | $p^a$ | OR(95%CI) | $p^b$ |
| 1 | A | G | A | A | 0.560 | 0.576 | 1 | - | 1 | - |
| 2 | G | G | C | G | 0.355 | 0.334 | 1.10 (0.85–1.42) | 0.48 | 1.09 (0.85–1.41) | 0.49 |
| 3 | A | A | A | A | 0.082 | 0.080 | 1.06 (0.68–1.64) | 0.80 | 1.06 (0.68–1.64) | 0.80 |
| rare | * | * | * | * | 0.003 | 0.010 | 0.48 (0.11–2.15) | 0.34 | 0.47 (0.11–2.13) | 0.33 |

Freq, frequency; ORs, odds ratios; CI, confidence interval.

[a]*p*-values were calculated using unconditional logistic regression analysis without adjustment.

[b]*p*-values were calculated using unconditional logistic regression analysis with an adjustment for age.

**Table 5. Functional annotation results.**

| SNP | NES | $p$-value | Tissue | Regulome DB Score | Function |
|---|---|---|---|---|---|
| rs1554286 | 0.30 | $2.2 \times 10^{-9}$ | Whole Blood | 1f | Likely to affect binding and linked to expression of a gene target |
| rs3024490 | 0.25 | $6.7 \times 10^{-5}$ | Nerve-Tibial | 6 | Minimal binding evidence |
| rs3024490 | 0.28 | $4.5 \times 10^{-9}$ | Whole Blood | | |
| rs1800871 | 0.25 | $7.3 \times 10^{-5}$ | Nerve-Tibial | 3a | Less likely to affect binding |
| rs1800871 | 0.28 | $4.5 \times 10^{-9}$ | Whole Blood | | |

NES, Normalized effect size; 1f, eQTL + TF binding / DNase peak; 3a, TF binding + any motif + DNase peak; 6, other.

## Discussion and conclusions

It is known that the polymorphisms of *IL-10* may contribute to the risk of breast cancer. Therefore, we selected six polymorphic sites that were located in the intron or promoter of *IL-10* and investigated their association with breast cancer susceptibility in Shaanxi Han women in China. The results show that the rs1554286 variant was associated with an increased risk of breast cancer in the co-dominant model. Additionally, rs3024490 and rs1800871 were associated with an increased breast cancer risk in co-dominant and dominant models. *In silico* analysis of the SNPs revealed that rs1554286, rs3024490, and rs1800871 may play important roles in the occurrence and development of breast cancer via regulating the expression of the target gene, *IL-10*. Our findings suggest that polymorphisms of *IL-10* may influence the risk of breast cancer among a group of Northwest Chinese Han women.

IL-10 is an important cytokine in immunity and cancer [19]. It is generally speculated that IL-10 plays a role in inhibiting the initiation and development of tumors by activating natural killer cells and cytotoxic T lymphocytes. Surviving tumor cells express IL-10, thus allowing them to escape immune recognition by reducing the expression of MHC class II [20–23]. A large number of association studies have found that polymorphisms of *IL-10* are involved in the susceptibility to many diseases, such as Bechcet's disease [24], systemic lupus erythematosus [25], and ulcerative colitis [26]. Meanwhile, some studies have indicated that IL-10 plays a vital role in the progression, invasion, migration, and growth of breast cancer [27, 28]. These results indicated that polymorphisms of *IL-10* have an effect on the development of breast

**Table 6. Breast cancer-associated single nucleotide polymorphisms (SNPs) with variants with $r^2 > 0.8$.**

| SNP | Chr: Position[a] | LD ($r^2$) | LD (D') | Allele[b] | Gene | HaploReg |
|---|---|---|---|---|---|---|
| rs1518111 | 1: 206771300 | 0.91 | 1 | C/T | *IL-10* (intron) | Promoter histone marks, Enhancer histone marks, DNAse, Proteins bound, Motifs changed, NHGRI/EBI GWAS hits, GRASP QTL hits, Selected eQTL hits |
| rs1518110 | 1: 206771516 | 0.89 | 0.98 | C/A | *IL-10* (intron) | Promoter histone marks, Enhancer histone marks, DNAse, Motifs changed, Selected eQTL hits |
| rs3024490 | 1: 206771966 | 1 | 1 | C/A | *IL-10* (intron) | Promoter histone marks, Enhancer histone marks, Motifs changed, GRASP QTL hits, Selected eQTL hits |
| rs1800872 | 1: 206773062 | 0.99 | 1 | G/T | *IL-10* (upstream) | Promoter histone marks, Enhancer histone marks, Proteins bound, Motifs changed, GRASP QTL hits, Selected eQTL hits |
| rs1800871 | 1: 206773289 | 1 | 1 | G/A | *IL-10* (upstream) | Promoter histone marks, Enhancer histone marks, DNAse, Motifs changed, NHGRI/EBI GWAS hits, Selected eQTL hits |

SNP, single nucleotide polymorphism; LD, linkage disequilibrium; eQTL, quantitative trait loci; GWAS, Genome-wide association study.

[a]Based on NCBI build 37 of the human genome.

[b]Reference allele/effect allele.

cancer. Further research is needed to elucidate the relationships among IL-10, tumorigenesis, and progression.

In our study, we found a correlation between the rs1554286, rs3024490, or rs1800871 and the risk of breast cancer in the Northwest Chinese Han women. Several studies have reported a relationship between *IL-10* polymorphisms and breast cancer susceptibility. Slattery et al. reported that allele "G" of rs1554286 and rs1800871 are associated with the risk of breast cancer among women with Native American ancestry [8]. However, a meta-analysis showed that the rs1800871 polymorphism had no relationship with breast cancer risk in Caucasians [29]. Our results were inconsistent with these findings. The rs1800871, which is located in the promoter, may alter the transcriptional regulation of miRNA on *IL-10*. Our functional prediction also suggested that these variants may influence the expression and transcription factor binding of *IL-10*.

No significant results were found for rs1518111, rs3021094, and rs3790622. A variant of rs1518111 has been suggested to correlate with the development of lymphedema following breast cancer treatment [30]. According to SNPInfo analysis, rs3021094 is likely to alter a putative transcription factor binding site for the transcription of *IL-10* [31], and this SNP was significantly associated with poorer overall survival in patients with colorectal cancer [32]. The haplotype "GACC" formed by rs1518111, rs3021094, rs3790622, and rs1800871 was significantly related to the risk of Bechet's disease [33]. However, to date, there is limited information on the association of these three SNPs and the risk of breast cancer. Future studies are needed to confirm our findings in a larger sample size.

Although our study was powered adequately, there were several limitations. First, our study focused on Chinese Han women who lived in Shaanxi province. Second, although we identified significant associations between six SNPs in *IL-10* and breast cancer susceptibility, the mechanisms responsible for these associations remain unclear. In the future, we will want to explore the specific factors which connect the SNPs and the occurrence and development of breast cancer, containing transcriptional regulator and signal pathway.

Overall, this study provided a valuable argument that *IL-10* may be a risk factor for the development and progression of breast cancer. Future studies should replicate these findings and involve women from other regions.

## Supporting information

**S1 Data.**
(XLSX)

## Acknowledgments

We thank all individuals for their participation.

## Author Contributions

**Conceptualization:** Miao Li, Xinhan Zhao.

**Data curation:** Miao Li, Chenli Yue, Guoquan Jin.

**Formal analysis:** Chenli Yue.

**Investigation:** Xiaoxiao Zuo.

**Methodology:** Miao Li, Xiaoxiao Zuo, Hulin Guo.

**Project administration:** Xinhan Zhao.

**Resources:** Guoquan Jin, Guanying Wang.

**Software:** Guanying Wang.

**Supervision:** Xinhan Zhao.

**Validation:** Fang Wu.

**Writing – original draft:** Miao Li, Shangke Huang.

**Writing – review & editing:** Xiaoxiao Zuo, Xinhan Zhao.

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
