## [Decision Letter · Decision Letter 0]

27 Dec 2019

PONE-D-19-31344

Interleukin-10 polymorphisms are risk factors for breast cancer susceptibility and functional annotation

PLOS ONE

Dear Dr. Zhao,

Thank you for submitting your manuscript to PLOS ONE. After careful consideration, we feel that it has merit but does not fully meet PLOS ONE’s publication criteria as it currently stands. Therefore, we invite you to submit a revised version of the manuscript that addresses the points raised during the review process and in particular points raised by Rev. 2.

We would appreciate receiving your revised manuscript by Feb 10 2020 11:59PM. To enhance the reproducibility of your results, we recommend that if applicable you deposit your laboratory protocols in protocols.io, where a protocol can be assigned its own identifier (DOI) such that it can be cited independently in the future. For instructions see: http://journals.plos.org/plosone/s/submission-guidelines#loc-laboratory-protocols

We look forward to receiving your revised manuscript.

Kind regards,

Giuseppe Novelli

Academic Editor

PLOS ONE

Journal Requirements:

2. In your Methods section, please provide additional information about the participant recruitment method and the demographic details of your participants. Please ensure you have provided sufficient details to replicate the analyses such as: a) the recruitment date range (month and year), b) a description of any inclusion/exclusion criteria that were applied to participant recruitment, c) a description of how participants were recruited, and d) descriptions of where participants were recruited.

3. Your ethics statement must appear in the Methods section of your manuscript. If your ethics statement is written in any section besides the Methods, please move it to the Methods section and delete it from any other section. Please also ensure that your ethics statement is included in your manuscript, as the ethics section of your online submission will not be published alongside your manuscript.

Reviewers' comments:

Reviewer's Responses to Questions

**Comments to the Author**

1. Is the manuscript technically sound, and do the data support the conclusions?

Reviewer #1: Yes

Reviewer #2: Partly

2. Has the statistical analysis been performed appropriately and rigorously? 

Reviewer #1: Yes

Reviewer #2: Yes

3. Have the authors made all data underlying the findings in their manuscript fully available?

Reviewer #1: Yes

Reviewer #2: Yes

4. Is the manuscript presented in an intelligible fashion and written in standard English?

Reviewer #1: Yes

Reviewer #2: Yes

5. Review Comments to the Author

Reviewer #1: The authors have selected six SNPs located in the intron or promoter of IL-10 and investigated their association with breast cancer susceptibility in Northwest Chinese Han women.

They have enrolled 530 breast cancer patients and 628 healthy individuals.

They results have shown that: 2 SNPs (rs3024490 and rs1800871) were associate with an increased breast cancer risk in co-dominant and dominant models.

The SNP, rs1554286, was significantly associated with an increased risk of breast cancer in the co-dominant model.

They reported the functional importance of these 3 SNPs that may influence the expression and transcription factor binding of IL-10.

The design of the study is appropriate for the question.

The methods are sufficiently explained and the results support the conclusions.

I have two remarks:

1) In the third line of page 12 you talk about the rs 1800871, as a mutation. It is a mistake.

2) You haven’t calculated if there are statistically significant differences between cases and controls for menopausal state.

Reviewer #2: In this paper the authors focused their attention on looking for an association between IL-10 single nucleotide polymorphisms (SNPs) and breast cancer susceptibility in Northwest Chinese Han women. The authors genotyped six SNPs from 530 breast cancer patients and 630 apparently healthy controls; the results were evaluated by unconditional logistic regression to assess the effective association between the considered polymorphisms and an increased breast cancer risk.

Two SNPs, rs3024490 and rs1800871, are reported to be significantly different between women with breast cancer and control cases, leading to an increased risk of breast cancer in dominant and co-dominant models. In addition, special attention was paid to a third SNP, rs1554286, which appears to be significantly associated with an increased breast cancer risk in a co-dominant model. Finally, functional evaluations suggested that these three variants could influence the expression and binding of IL-10 transcription factors.

The most relevant flaws that we found in the study are:

• The clinical validity of the OR depends on how the control group is representative of the entire population and how influent are all the other factors present in the cases, which could independently give an association with the disease. Given the lack of representativeness of the sample, as it relates only to the province of Shaanxi, it could be useful to report the values of likelihood ratio which take into account the epidemiological data of this specific province. This allows to be clearer about the risk of bias due to a stratification of the population;

• In order to support the hypothesis, that the presence of polymorphisms affects the expression of IL-10, it would be appropriate to report data that support the formulation of this hypothesis (e.g. variation of IL-10 transcripts in the presence of a particular allele or haplotype), taking in account that the haplotypes analyzed do not show a significant association with breast cancer, as reported by the same authors;

• When so few SNPs are genotyped, it is difficult to establish the failure rate and heterozygosity rate for markers quality control procedures. It is however advisable to remove individuals with low call rates. This article does is not explicit on the call rate of individuals, neither on the threshold at which individuals are excluded, or whether individuals have been excluded. We believe that these data should be highlighted because they define the reproducibility of the experiment. There are also no references about the presence of any familiar relationship between the selected individuals, which might be source of bias in a population-based case-control study, given that the enrollment is limited within a specific province;

• The authors speculated that the lower frequency allele could be associated with the disease. However, in table 2, when the frequencies of the minor allele are compared in the two groups of cases and controls, we notice that there is no significant difference (p-values > 0.05) as reported same authors within the text; we suggest reporting the reasons that led the authors to consider the minor allele as the one associated with the disease and also the allelic frequencies of the major allele;

• We suggest the authors to provide more clarifications on the selection criteria of the 6 starting SNPs of the study;

• For the benefit of the reader, it could be useful to specify the criteria on the basis of which an "age correction" of the subjects in the study was made;

• Table 3 is not clear in the graphic construction and difficult to read, especially the column related to the genotypes referred to each model (describe all the genotypes in comparison for each model) and in the section related to the p-value (review text alignment and report all p-values);

• At first glance, the title seems to make a statement that is not sufficiently proven by the results drawn in the article, if not by hypothetical models which, however, are not sufficiently validated. We therefore consider premature to state that "IL-10 polymorphisms ARE risk factors for breast cancer susceptibility". It would be advisable to make a revision of the title according to the hypothetical value of the results;

• Finally, we advise the authors to avoid elements of repetitiveness in the discussion, in order to make reading more smooth and pleasant.

6. PLOS authors have the option to publish the peer review history of their article (what does this mean?). If published, this will include your full peer review and any attached files.

Reviewer #1: No

Reviewer #2: No

---

## [Author Response · Author response to Decision Letter 0]

12 Mar 2020

Dear editors and reviewers: 

On behalf of my co-authors, we thank you very much for giving us an opportunity to revise our manuscript, we appreciate you and reviewers very much for your positive and constructive comments on our manuscript entitled “Interleukin-10 polymorphisms are risk factors for breast cancer susceptibility and functional annotation ” (ID: PONE-D-19-31344 R1). Those comments are all valuable and helpful for revising and improving our paper, as well as the important guiding significance to our researcher. We have studied comments carefully and have made correction which we hope meet with approval. We hope meet with approval.

We have carefully evaluated the reviewers’ critical comments and thoughtful suggestions, responded to these suggestions point-by-point, and revised the manuscript. Revised portion are marked using the track changes mode in the marked manuscript.

The main corrections in the paper and the responds to the reviewer’s comments are as follows:

Response to Journal Requirements:

1.When submitting your revision, we need you to address these additional requirements. Please ensure that your manuscript meets PLOS ONE's style requirements, including those for file naming. The PLOS ONE style templates can be found at http://www.journals.plos.org/plosone/s/file?id=wjVg/PLOSOne_formatting_sample_main_body.pdf and http://www.journals.plos.org/plosone/s/file?id=ba62/PLOSOne_formatting_sample_title_authors_affiliations.pdf

Response: We thank the editor's for scrutinize. We revised the manuscript according to the reviewer's suggestion. Thank you again for your careful review of our manuscript.

2.In your Methods section, please provide additional information about the participant recruitment method and the demographic details of your participants. Please ensure you have provided sufficient details to replicate the analyses such as: a) the recruitment date range (month and year), b) a description of any inclusion/exclusion criteria that were applied to participant recruitment, c) a description of how participants were recruited, and d) descriptions of where participants were recruited.

Response: We thank the editor's for scrutinize. We have revised this issue in the manuscript according to the reviewer's suggestion (line 92-102, page 15). Thank you again for your careful review of our manuscript.

3. Your ethics statement must appear in the Methods section of your manuscript. If your ethics statement is written in any section besides the Methods, please move it to the Methods section and delete it from any other section. Please also ensure that your ethics statement is included in your manuscript, as the ethics section of your online submission will not be published alongside your manuscript.

Response: We thank the editor's for scrutinize. We have revised this issue in the manuscript according to the reviewer's suggestion (line 83-90, page 5). Thank you again for your careful review of our manuscript.

Responds to the reviewer’s comments:

Response to Reviewer #1:

Major Concerns:

1)In the third line of page 12 you talk about the rs 1800871, as a mutation. It is a mistake.

Response: We thank the reviewer’s for scrutinize. I am sorry for this type of writing error in our manuscript. And we have revised such mistakes in the manuscript according to the reviewer's suggestion (line 3, page 12). Thank you again for your careful review of our manuscript.

2) You haven’t calculated if there are statistically significant differences between cases and controls for menopausal state.

Response: We thank the reviewer’s for scrutinize. The menopause information of the case group in this study is comprehensive, while the menopause-related information of the healthy control group is not comprehensive, so we have not calculated whether there is a statistically significant difference between the menopausal case and the control group. When we conduct further in-depth research in the future, we will collect the clinical information of the subjects as much as possible, and conduct a more comprehensive and in-depth experimental design. Thank you again for your careful review of our manuscript.

Response to Reviewer #2:

1.The clinical validity of the OR depends on how the control group is representative of the entire population and how influent are all the other factors present in the cases, which could independently give an association with the disease. Given the lack of representativeness of the sample, as it relates only to the province of Shaanxi, it could be useful to report the values of likelihood ratio which take into account the epidemiological data of this specific province. This allows to be clearer about the risk of bias due to a stratification of the population;

Response: We thank the reviewer’s for scrutinize. The cases and control population in this study were both Han populations from Shaanxi Province. We revised the writing in the manuscript based on the reviewer's suggestion. Thank you again for your careful review of our manuscript.

2.In order to support the hypothesis, that the presence of polymorphisms affects the expression of IL-10, it would be appropriate to report data that support the formulation of this hypothesis (e.g. variation of IL-10 transcripts in the presence of a particular allele or haplotype), taking in account that the haplotypes analyzed do not show a significant association with breast cancer, as reported by the same authors;

Response: We thank the reviewer’s for scrutinize. At present, many studies have reported that changes in SNPs loci can affect the expression of regulatory genes in a variety of ways (1. Cibele Masotti, Lucia M. Armelin-Correa, Alessandra Splendore,等. A functional SNP in the promoter region of TCOF1 is associated with reduced gene expression and YY1 DNA–protein interaction[J]. 359(none):0-52. 2. Yuan L, Shen S, Luo C. SNP Identification in α_(2A)-Adrenergic Receptor Gene in Chinese and the Effect on Gene Expression[J]. 2003, 17(6):277-282. 3.Safaa I. Tayel, Eman A. M. Fouda, Elsayed I. Elshayeb,ect. Biochemical and molecular study on interleukin‐1β gene expression and relation of single nucleotide polymorphism in promoter region with Type 2 diabetes mellitus[J]. Journal of Cellular Biochemistry, 2018, 119(7)).

In this study, we referenced reference “Xu Z , Taylor J A . SNPinfo: integrating GWAS and candidate gene information into functional SNP selection for genetic association studies[J]. Nucleic Acids Research, 2009, 37(Web Server):W600-W605”, and initially annotated the functions of selected SNPs through the GTEx database and the HaploReg database. The prediction results showed that these sites have different regulatory functions ((line 189-205, page 10). However, further research is needed to further explore how the candidate SNPs affect IL-10 expression. We will further collect samples to explore this issue later. Thank you again for your careful review of our manuscript.

3.When so few SNPs are genotyped, it is difficult to establish the failure rate and heterozygosity rate for markers quality control procedures. It is however advisable to remove individuals with low call rates. This article does is not explicit on the call rate of individuals, neither on the threshold at which individuals are excluded, or whether individuals have been excluded. We believe that these data should be highlighted because they define the reproducibility of the experiment. There are also no references about the presence of any familiar relationship between the selected individuals, which might be source of bias in a population-based case-control study, given that the enrollment is limited within a specific province;

Response: We thank the reviewers for their review. We modified the manuscript based on the reviewer's recommendations (page 111-116, line 6; page 127, line 7; page 102, line 6). Thank you again for your careful review of our manuscript.

4.The authors speculated that the lower frequency allele could be associated with the disease. However, in table 2, when the frequencies of the minor allele are compared in the two groups of cases and controls, we notice that there is no significant difference (p-values > 0.05) as reported same authors within the text; we suggest reporting the reasons that led the authors to consider the minor allele as the one associated with the disease and also the allelic frequencies of the major allele;

Response: We thank the reviewer’s for scrutinize. As shown in Table 2, We hypothesized that the minor allele of the SNP was the risk allele to analyze the correlation with breast cancer susceptibility. Our results show that under this allele model, the minor alleles of all candidate SNPs have no significant correlation with breast cancer risk (p> 0.05).

However, the genetic loci in nature exist in the form of genotypes. We further analyzed the relationship between the corresponding genotypes and the risk of breast cancer under different genetic models in Table 3. 

5.We suggest the authors to provide more clarifications on the selection criteria of the 6 starting SNPs of the study;

Response: We thank the reviewer’s for scrutinize. In this study, six SNPs (rs1554286, rs1518111, rs3021094, rs3790622, rs3024490, rs1800871) of IL-10 were selected from the DbSNP (http://www.hapmap.org/index.html.en) and SNP Consortium (http://snp.cshl.org/) databases based on the following criteria. We took allele frequency into consideration during genotyping. The lower frequency alleles were coded as minor alleles. All SNPs were selected at a minor allele frequency >5% in 1,000 genome project (http://www.internationalgenome.org/). In addition, the genotype distribution of SNPs in the control group was in accordance with Hardy-weinberg equilibrium (HWE) (p > 0.05). The genotyping of Agena MassARRAY RS1000 for these SNPs exceeded 95.0%. The Haploview software package (version 4.2) was used to estimate the pairwise linkage disequilibrium (LD) of the IL-10 polymorphic site. When r2 (measured value of LD) > 0.8, SNP can represent all polymorphisms in a block (page 118-127, line 7).

6.For the benefit of the reader, it could be useful to specify the criteria on the basis of which an "age correction" of the subjects in the study was made;

Response: We thank the reviewer’s for scrutinize. As shown in Table 1, the average age of cases and controls was 50.69 years and 51.04 years, respectively. Welch's t-test showed a significant difference in age between the cases and the controls (p = 0.001). In order to exclude the effects of age differences and confounding factors in subsequent statistics, we adjusted the age.

7.Table 3 is not clear in the graphic construction and difficult to read, especially the column related to the genotypes referred to each model (describe all the genotypes in comparison for each model) and in the section related to the p-value (review text alignment and report all p-values);

Response: We thank the reviewer’s for scrutinize. For a single nucleotide polymorphism, we hypothesized that the minor allele of each SNP was a risk factor compared with the wild-type allele. In this case-control study, we referred to a published article entitled “Basic statistical analysis in genetic case-control studies” about statistical analysis (Clarke G M, Anderson C A, Pettersson F H, et al. Basic statistical analysis in genetic case-control studies [J]. Nature Protocol, 2011, 6(2):121-133). In this paper, the author showed that the disease risk is really the cumulative effect of the risk alleles. 

When we consider a genetic marker consisting of a single bi-allelic locus with alleles a and A (i.e., a SNP), three possible genotypes are G/G, G/A and A/A (A is the minor allele). Multiple genetic models are shown as follows: allelic model: G vs A; dominant model: AA+AG vs GG; recessive model: AA vs AG+GG; codominant model: AA vs AG vs GG; the log-additive model indicates that the risk of disease is increased 1-fold for genotype a/A and by 2-fold for genotype A/A. The corresponding P values correspond to different models.

For example, in Table 3, rs1554286, Co-dominant refers to G / G vs A / G vs A / A, and the corresponding p-values before and after correction are 0.018 * and 0.018 *, respectively; Dominant model refers to A / G-G / G vs A / A, and the corresponding p-values are 0.081 and 0.084 before and after correction, respectively; Recessive model refers to G / G vs A / A-A / G, and the corresponding p-values before and after correction are 0.11 and 0.11, respectively; the corresponding p-values before and after correction in the Log-additive model are 0.60 and 0.62, respectively. We modified the Table 3 based on the reviewer's recommendations. Thank you again for your careful review of our manuscript.

8.At first glance, the title seems to make a statement that is not sufficiently proven by the results drawn in the article, if not by hypothetical models which, however, are not sufficiently validated. We therefore consider premature to state that "IL-10 polymorphisms ARE risk factors for breast cancer susceptibility". It would be advisable to make a revision of the title according to the hypothetical value of the results;

Response: We thank the reviewer’s for scrutinize. We modified the manuscript based on the reviewer's recommendations (page 1, line 1). Thank you again for your careful review of our manuscript.

9.Finally, we advise the authors to avoid elements of repetitiveness in the discussion, in order to make reading more smooth and pleasant.

Response: We thank the reviewer’s for scrutinize. We modified the manuscript based on the reviewer's recommendations (page 226, line 12). Thank you again for your careful review of our manuscript.

We appreciate for Editors/Reviewers’ warm work earnestly, the reviewer’s comments are quite helpful. We revised our paper point-by-point problems. Finally, we hope that the correction will meet with approval. Looking forward to hearing from you. 

Thank you and best regards!

Correspondence author:

Xinhan Zhao

Tel: +86-13909240672

E-mail: zhaohanxin21@163.com

Address: #277 West Yanta Road, Xi’an 710061, Shaanxi, China

---

## [Editor Report · Decision Letter 1]

9 Apr 2020

The effect of Interleukin 10 Polymorphisms  on Breast Cancer Susceptibility in Han Women in Shaanxi Province

PONE-D-19-31344R1

Dear Dr. Zhao,

We are pleased to inform you that your manuscript has been judged scientifically suitable for publication and will be formally accepted for publication once it complies with all outstanding technical requirements.

With kind regards,

Giuseppe Novelli

Academic Editor

PLOS ONE
---

## [Editor Report · Acceptance letter]

28 Apr 2020

PONE-D-19-31344R1 

The effect of Interleukin 10 Polymorphisms on Breast Cancer Susceptibility in Han Women in Shaanxi Province 

Dear Dr. Zhao:

I am pleased to inform you that your manuscript has been deemed suitable for publication in PLOS ONE. Congratulations! Your manuscript is now with our production department. 

With kind regards,

on behalf of

Prof. Giuseppe Novelli 

Academic Editor

PLOS ONE